# Genetic Parameters of Reproductive Performances in Hungarian Large White, Landrace, and Their Crossbred F1 Pigs from 2010 to 2018

Oleksandr Kodak *, Henrietta Nagyne-Kiszlinger, Janos Farkas, György Köver and Istvan Nagy 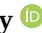

Kaposvár Campus, Hungarian University of Agriculture and Life Sciences, Guba Sándor Str. 40., H-7400 Kaposvár, Hungary
* Correspondence: oleksandr.kodak1984@gmail.com; Tel.: +36-70-351-4504 or +38-050-506-98-31

**Abstract:** Genetic parameters, breeding values, and aggregate breeding values of number of piglets born alive (NBA), number of weaned piglets (NWE), and litter weight at weaning (LWWE) were predicted in the Hungarian Large White, Hungarian Landrace breeds, and in their cross (F1). Seven repeatability animal models were used. BLUP and REML methodology were used to estimate breeding values and variance–covariance components. PEST and VCE 6 software were used for estimating breeding values and variance components. Heritability for NBA and NWE was the same for all seven models. On the contrary, heritability estimates for LWWE were higher in comparison with NBA and NWE. The permanent environmental variance component was small for all traits. The large White breed had positive and significant genetic trends for all seven models and for all three traits. Landrace breed had significant trends for NBA, which was negative, and for NWE, the results were positive. The constructed indices result in one number (i.e., aggregate genetic merit); thus, the animals can be selected based on their overall performance considering the various aspects.

**Keywords:** pig; breeding; diversity; desired gain index; aggregate genotype

## 1. Introduction

The various methods of quantitative genetics have been applied by breeders for decades in order to improve the performance of domesticated animals in some pre-defined traits. In the pig breeding sector, similarly to other multiparous species, the crossing is widely used in order to increase performance, where generally separate breeds are used to increase reproductive (e.g., number of piglets born alive, number of piglets born dead, number of piglets weaned, litter weight at weaning) and growth and carcass (e.g., average daily gain, feed conversion ratio, lean meat percentage, the proportion of valuable cuts) performances [1]. Looking at the various traits used in pig breeding number of piglets born alive is among the most important trait from the economic viewpoint, as this trait has the largest economic weight besides the feed conversion ratio [2,3]. Monitoring the actual genetic parameters of the economically important traits and evaluating the genetic progress of the populations is always an important task that helps us to determine the expected efficiency of the breeding program. The main objective of our current study was to evaluate the genetic merit of several reproductive traits (i.e., number of piglets born alive, number of weaned piglets, and litter weight at weaning) in two pure breeds and their cross because we think that the viability of newborn generations of pigs or other animals is a guarantee for the future survival of a species or breed. We used in our current research a classical approach for evaluating genetic parameters by applying the BLUP methodology [4]. The expected selection response is a guarantee for obtaining the properly aimed population inside species. We also speak of an artificially controlled environment in our case selection process.

To characterize and possibly predict the response, we built and used linear regressions based on available genetic parameters for attaining the genetic trends for two pure breeds and their cross [5].

Additionally, based on these predictions, the overall genetic merit was also determined as an inevitable condition of carrying out efficient selection using the procedure of desired gain [6,7]. Coefficients of the index for breeding value estimation were realized for targeted traits by applying aggregate genotype where estimated index-weighted factors were taken into account.

## 2. Materials and Methods

### 2.1. Data Collection

In Hungary, the selection process and prediction of genetic merit for reproduction traits for maternal breeds are focused mainly on the number of piglets born alive and litter weight at weaning. Data used in our research were collected by Hungarian Pig Breeders' Association from 56 herds between 2010 and 2018 in the course of the field test. Farrowing data of Hungarian Large White (HLW), Hungarian Landrace (HLR), and F1 generation (Large White boars mated with Landrace sows and Landrace boars mated with Large White sows) were collected. The recorded traits were piglets born alive (NBA), number of weaned piglets (NWE), and litter weight at weaning (LWWE), which directly affect future genetic diversity for any species. Descriptive statistics of the farrowing records, herds, and sows are summarized in Table 1.

**Table 1.** Farrowing information 2010–2018.

| Breed/Name | Herds | Sows | Number of Farrowings |
|:---:|:---:|:---:|:---:|
| Total | 56 | 27,561 | 73,871 |
| Large White | 42 | 16,749 | 50,147 |
| Landrace | 23 | 4372 | 12,645 |
| F1 | 34 | 6440 | 11,079 |

F1—cross of Hungarian Large White and Hungarian Landrace.

### 2.2. Animal Model

The farrowing data set was used for testing 7 repeatability animal models to select the most appropriate model for estimating genetic parameters, and the structures of these models are presented in Table 2. The breeds (HLW and HLR) and their cross F1) were analyzed together as one population constructing their common pedigree. In this way, the heterosis effect could also be accounted for, including the breed/construction of the animals in every model. Traits that were analyzed in models are the number of piglets born alive (NBA), number of weaned piglets (NWE), and litter weight at weaning (LWWE). Best Linear Unbiased Prediction (BLUP) and Restricted maximum likelihood methodology(REML) were used for the estimation of breeding values and variance–covariance components. PEST [8] and VCE 6 [9] software were used (for data coding) for the estimation of breeding values and variance components.

The basic repeatability model was:

$$y = Xb + Za + Wpe + e$$

where y is the vector of observations; b is the vector of fixed effects; a is the vector of random animal effects; pe is the random vector of permanent environmental effects (dam identity); e is the vector of random residual effects; and X, Z, and W are the incidence matrices relating records to fixed, animal, and random maternal permanent effects, respectively.

**Table 2.** The structure of the applied animal models.

| Model | Traits | | | Factors (Type) | | | | | | | | |
|---|---|---|---|---|---|---|---|---|---|---|---|---|
| | | | | Animal | Rep | FSA | SWA | FYM | WYM | Breed | Herd | Parity |
| | NBA | NWE | LWWE | (A) | (R) | (C) | (C) | (F) | (F) | (F) | (F) | (F) |
| 1 | x | x | | x | x | | | x | x | x | x | x |
| 2 | x | x | | x | x | x | x | x | x | x | x | |
| 3 | x | x | x | x | x | | | x | x | x | x | x |
| 4 | x | x | x | x | x | x | x | x | x | x | x | |
| 5 | x | | x | x | x | | | x | x | x | x | x |
| 6 | x | | x | x | x | x | x | x | x | x | x | |
| 7 | | x | x | x | x | | | x | x | x | x | x |

NBA: number of piglets born alive; NWE: number of weaned piglets; LWWE: litter weight at weaning; REP: repeatability measurements; FSA: age of farrowing sows; SWA: sows age at weaning; FYM: farrowing year-month; WYM: weaning year–month; A: additive genetic effect; R: random effect; F: fixed effect.

Expected values of a, c and e were E(a) = E(c) = E(e) = 0. The variance–covariance structure was assumed to be $V(a) = A\sigma^2a$, $V(c) = I\sigma^2c$ $V(e) = I\sigma^2e$ and $V(a) = A\sigma^2a$, $V(e) = I\sigma^2e$ $Cov(a,e) = Cov(e,a) = Cov(c,e) = Cov(e,c) = 0$ and $Cov(a,e) = Cov(e,a) = 0$, where A is the numerator relationship matrix. Additionally, $cov(y,a) = ZAI\sigma^2a$.

The suitability of the different models was compared using the log-likelihood values calculated by the VCE software. The model with the largest log-likelihood value provides the best fit.

SAS 9.4 [10] was used for descriptive statistical analysis, which is summarized in Table 3. In addition, SAS 9.4 was also applied to calculating the genetic trend for every trait, which is the linear regression coefficient of the average breeding value of animals born in the same year (regressed on the successive years of birth). Additionally, Mix software was used [11] for the calculation of the maternal desired index with the purpose of improving all traits by one additive standard deviation where the assignment of relative economic values of the examined traits is not necessary. The breeding goals are defined as the ultimate levels of the traits of interest. The desired gain index is constructed to attain the predetermined breeding goals in minimum number of generations of selection. A detailed description of the index weighing factors' calculation was given by Yamada et al. [12]. The calculated index scores were transformed in order to obtain index mean and standard deviation equal to 100 and 20, respectively, as it is used in Hungarian pig breeding [13].

**Table 3.** Descriptive statistics of the measured traits.

| Group | Trait | Mean | SD | Maximum | Minimum |
|---|---|---|---|---|---|
| Large White | NBA | 11.23 | 2.81 | 19 | 1 |
| | NWE | 10.28 | 1.94 | 16 | 1 |
| | LWWE | 75.50 | 19.27 | 130 | 5 |
| Landrace | NBA | 11.03 | 2.59 | 19 | 1 |
| | NWE | 10.34 | 1.71 | 16 | 1 |
| | LWWE | 69.76 | 15.38 | 130 | 6 |
| F1 | NBA | 11.16 | 2.80 | 19 | 1 |
| | NWE | 10.23 | 1.58 | 16 | 1 |
| | LWWE | 78.98 | 18.28 | 130 | 6 |

F1—cross of Hungarian Large White and Hungarian Landrace; NBA: number of piglets born alive; NWE: number of weaned piglets; LWWE: litter weight at weaning; SD: standard variation.

## 3. Results

### 3.1. Models' Fit, Estimated Genetic Parameters of the Best Fitting Model

In Table 2, it can be seen that the models differed in one fixed factor, including either parity or the age of the sow at farrowing and/or at weaning. Based on the estimated log-likelihood-value models (Table 4), including age were inferior compared to those models containing parity both for the two-trait (154 325 vs. 156 095; 145 979 vs. 148 224) and for the three-trait (167 132 vs. 169 370) models. Altogether the best-fitting model was model 7, with the lowest log-likelihood value of 95 842. On the contrary, model 4 gave the smallest fit with a log-likelihood value of 169 370 (Table 4). Estimated variance–covariance components of the best-fitting model (model 4) are given in Table 4.

**Table 4.** Log-likelihood values of the used 7 models and estimated variances (diagonals) and covariances (off-diagonals) of the best fitting model (model 7).

| Trait | | | NBA | NWE | LWWE |
|---|---|---|---|---|---|
| Type | | A | | 0.20 | 1.45 |
| Models/log likelihood down | | | | | 31.1 |
| 1 | 154 325 | | | | |
| 2 | 156 095 | Pe | | 0.06 | 0.56 |
| 3 | 167 132 | | | | 5.66 |
| 4 | 169 370 | | | | |
| 5 | 145 979 | Res | | 2.59 | 16.9 |
| 6 | 148 224 | | | | 203.8 |
| 7 | 95 842 | | | | |

A: additive genetic; Pe: maternal permanent; Res: residual; NBA: number of piglets born alive; NWE: number of weaned piglets; LWWE: litter weight at weaning.

### 3.2. Heritability, Maternal Permanent Effect, and Genetic Trends

Heritability and maternal permanent effects were presented, characterizing all three genotypes. Heritability and its standard errors are presented in Table 5. Heritability for NBA and NWE was low for all 7 models ranging between 0.07 and 0.08 and between 0.06 and 0.07, respectively. On the contrary, the heritability of LWWE was higher in comparison with NBA, showing heritability estimates between 0.12 and 0.14.

**Table 5.** Estimated heritabilities and maternal permanent effect of the examined traits.

| Model | $h^2$ | | | Pe | | |
|---|---|---|---|---|---|---|
| | NBA | NWE | LWWE | NBA | NWE | LWWE |
| 1 | 0.08 ± 0.004 | 0.07 ± 0.004 | | 0.07 ± 0.004 | 0.02 ± 0.003 | |
| 2 | 0.08 ± 0.004 | 0.07 ± 0.005 | | 0.06 ± 0.004 | 0.01 ± 0.004 | |
| 3 | 0.07 ± 0.003 | 0.07 ± 0.003 | 0.13 ± 0.004 | 0.07 ± 0.003 | 0.02 ± 0.003 | 0.02 ± 0.003 |
| 4 | 0.08 ± 0.002 | 0.06 ± 0.003 | 0.12 ± 0.004 | 0.06 ± 0.002 | 0.02 ± 0.004 | 0.02 ± 0.003 |
| 5 | 0.08 ± 0.005 | | 0.14 ± 0.005 | 0.07 ± 0.005 | | 0.02 ± 0.005 |
| 6 | 0.08 ± 0.005 | | 0.14 ± 0.005 | 0.06 ± 0.005 | | 0.01 ± 0.004 |
| 7 | | 0.07 ± 0.004 | 0.13 ± 0.005 | | 0.02 ± 0.004 | 0.02 ± 0.005 |

NBA: number of piglets born alive; NWE: number of weaned piglets; LWWE: litter weight at weaning; h2: heritability; Pe: maternal permanent effect.

Maternal permanent effects are presented in Table 5, and the values are low for all traits. For NBA, NWE, and LWWE, it ranged between 0.06 and 0.07, between 0.01 and 0.02, and between 0.01 and 0.02, respectively.

In Table 6 (HLW), Table 7 (HLR), and Table 8 (F1) are summarized the estimated genetic trends for all seven models. The HLW had positive and significant trends for all seven models and for all three traits. Thus, NBA had a prediction ranging between 0.04 and 0.05 piglets per year, NWE had a lower value contrary to NBA ranging between 0.01 and 0.02 piglets per year, and the LWWE trend was between 0.08 and 0.1 kg per year.

**Table 6.** Estimated trends for Large White.

| Models | NBA | | NWE | | LWWE | |
|---|---|---|---|---|---|---|
| | Pr > |t| | B | Pr > |t| | B | Pr > |t| | B |
| 1 | <0.0001 | 0.04 | <0.0001 | 0.02 | | |
| 2 | <0.0001 | 0.04 | <0.0001 | 0.02 | | |
| 3 | <0.0001 | 0.04 | <0.0001 | 0.01 | <0.0001 | 0.09 |
| 4 | <0.0001 | 0.05 | <0.0001 | 0.02 | <0.0001 | 0.1 |
| 5 | <0.0001 | 0.04 | | | <0.0001 | 0.1 |
| 6 | <0.0001 | 0.05 | | | <0.0001 | 0.1 |
| 7 | | | <0.0001 | 0.01 | <0.0001 | 0.08 |

NBA: number of piglets born alive; NWE: number of weaned piglets; LWWE: litter weight at weaning.

**Table 7.** Estimated trends for Landrace.

| Models | NBA | | NWE | | LWWE | |
|---|---|---|---|---|---|---|
| | Pr > |t| | B | Pr > |t| | B | Pr > |t| | B |
| 1 | <0.0001 | −0.02 | 0.04 | 0.003 | | |
| 2 | <0.0001 | −0.01 | 0.0007 | 0.005 | | |
| 3 | <0.0001 | −0.02 | 0.005 | 0.004 | 0.5 | −0.02 |
| 4 | <0.0001 | −0.02 | <0.0001 | 0.006 | 0.3 | 0.02 |
| 5 | <0.0001 | −0.02 | | | 0.7 | −0.009 |
| 6 | <0.0001 | −0.01 | | | 0.1 | 0.03 |
| 7 | | | 0.001 | 0.005 | 0.6 | −0.01 |

NBA: number of piglets born alive; NWE: number of weaned piglets; LWWE: litter weight at weaning.

**Table 8.** Estimated trends for F1.

| Models | NBA | | NWE | | LWWE | |
|---|---|---|---|---|---|---|
| | Pr > |t| | B | Pr > |t| | B | Pr > |t| | B |
| 1 | <0.0001 | −0.02 | <0.0001 | −0.02 | | |
| 2 | <0.0001 | −0.02 | <0.0001 | −0.02 | | |
| 3 | <0.0001 | −0.02 | <0.0001 | −0.02 | 0.004 | −0.06 |
| 4 | <0.0001 | −0.01 | <0.0001 | −0.01 | 0.2 | −0.02 |
| 5 | <0.0001 | −0.02 | | | <0.0001 | −0.08 |
| 6 | <0.0001 | −0.01 | | | 0.007 | −0.05 |
| 7 | | | <0.0001 | −0.02 | 0.002 | −0.06 |

NBA: number of piglets born alive; NWE: number of weaned piglets; LWWE: litter weight at weaning.

The HLR (Table 7) had significant genetic trends for NBA, showing $-0.01--0.02$ piglets per year, and for NWE ranging between 0.003 and 0.006 piglets per year. Genetic trends of LWWE in the case of Landrace were not significant.

The F1 (Table 8) had significant genetic trends for all three traits but with negative tendencies. NBA and NWE showed similar tendencies with $-0.01--0.02$ and $-0.01--0.02$ piglets per year, respectively. The genetic trend for LWWE was not significant in model number 4, but for all other models, we observed negative trends of $-0.02--0.08$ kg per year.

### 3.3. Index Weighting Factors, Genetic Correlation, and Index by Breeds

The calculated index scores are presented in Table 9. The calculated maternal index can directly be compared with the index, which is used by the Hungarian Pig Breeders Association [11]. Additionally, the index of the Hungarian Pig Breeders Association was constructed based on the calculation of the economic values, but indexes considered in our research were based on so-called desired gains improving every trait with one additive standard deviation.

**Table 9.** Selection indexes weighting factors for the examined traits.

| Models | Index |
|--------|-------|
| 1 | $15.6805 \times ebv1 + 26.3388 \times ebv2$ |
| 2 | $15.1689 \times ebv1 + 25.9797 \times ebv2$ |
| 3 | $14.4687 \times ebv1 + 11.2599 \times ebv2 + 1.7521 \times ebv3$ |
| 4 | $13.8544 \times ebv1 + 12.3412 \times ebv2 + 1.8160 \times ebv3$ |
| 5 | $16.8858 \times ebv1 + 2.1646 \times ebv3$ |
| 6 | $16.2625 \times ebv1 + 2.1174 \times ebv3$ |
| 7 | $25.2420 \times ebv2 + 2.0141 \times ebv3$ |

ebv are the estimated breeding values for traits: ebv1—number of piglets born alive, ebv2—number of weaned piglets, and ebv3—litter weight at weaning.

In Table 10 (HLW), Table 11 (HLR), and Table 12 (F1) are summarized in the genetic correlations between index and traits. The index for HLW (Table 10) had a high correlation with traits (from 0.74 to 0.91). NWE had a moderate correlation with NBA (ranging from 0.48 to 0.50). On the contrary, LWWE had a low to moderate correlation with NBA and NWE (ranging from 0.32 to 0.37) and (ranging from 0.56 to 0.58), respectively.

HLR (Table 11) index had a moderate to high correlation with traits (0.67–0.91). NBA and NWE had a moderate correlation (0.56–0.60), but LWWE had a low to moderate correlation with NBA and NWE (0.33–0.39 and 0.28–0.31, respectively).

F1 index (Table 12) had a moderate to high correlation with traits (0.66–0.89). NBA and NWE had a moderate correlation (0.38–0.44); in contrast, LWWE had a low to moderate correlation with NBA and NWE (0.20–0.24 and 0.38–0.43, respectively).

In Table 13, genetic correlations are shown between the pure-bred, the cross-bred performances for two pure breeds, and their cross simultaneously for all seven models. We found low to moderate correlations between the traits. Thus, NBA correlated with NWE and LWWE in a range of 0.49–0.52 and 0.30–0.32, respectively. The correlation between NWE and LWWE ranged between 0.56 and 0.59.

In Table 14, the results of index scores are summarized, which are presented for seven models and every genotype separately. As we can see, results for HLW wider range of index scores were found in all seven models compared to HLR and F1.

**Table 10.** Genetic correlation coefficient (Pearson's correlation) between examined traits and the selection index scores for the HLW (Pr > |t| < 0.0001).

| Models | Traits | | | |
| | Traits | NWE | LWWE | Index |
|---|---|---|---|---|
| 1 | NBA | 0.48 | | 0.85 |
| | NWE | | | 0.88 |
| 2 | NBA | 0.50 | | 0.86 |
| | NWE | | | 0.88 |
| 3 | NBA | 0.48 | 0.32 | 0.74 |
| | NWE | | 0.58 | 0.78 |
| | LWWE | | | 0.85 |
| 4 | NBA | 0.48 | 0.33 | 0.74 |
| | NWE | | 0.57 | 0.77 |
| | LWWE | | | 0.86 |
| 5 | NBA | | 0.32 | 0.74 |
| | LWWE | | | 0.87 |
| 6 | NBA | | 0.37 | 0.77 |
| | LWWE | | | 0.88 |
| 7 | NWE | | 0.56 | 0.86 |
| | LWWE | | | 0.91 |

NBA: number of piglets born alive; NWE: number of weaned piglets; LWWE: litter weight at weaning.

**Table 11.** Genetic correlation coefficient (Pearson's correlation) between examined traits and the selection index score for the HLR (Pr > |t| < 0.0001).

| Models | Traits | | | |
| | Traits | NWE | LWWE | Index |
|---|---|---|---|---|
| 1 | NBA | 0.59 | | 0.90 |
| | NWE | | | 0.88 |
| 2 | NBA | 0.60 | | 0.91 |
| | NWE | | | 0.88 |
| 3 | NBA | 0.56 | 0.34 | 0.83 |
| | NWE | | 0.31 | 0.68 |
| | LWWE | | | 0.77 |
| 4 | NBA | 0.56 | 0.33 | 0.83 |
| | NWE | | 0.28 | 0.67 |
| | LWWE | | | 0.77 |
| 5 | NBA | | 0.36 | 0.83 |
| | LWWE | | | 0.82 |
| 6 | NBA | | 0.39 | 0.83 |
| | LWWE | | | 0.83 |
| 7 | NWE | | 0.28 | 0.76 |
| | LWWE | | | 0.84 |

NBA: number of piglets born alive; NWE: number of weaned piglets; LWWE: litter weight at weaning.

**Table 12.** Genetic correlation coefficient (Pearson's correlation) between examined traits and the selection index for F1 (Pr > |t| < 0.0001).

| Models | Traits | | | |
|---|---|---|---|---|
| | **Traits** | **NWE** | **LWWE** | **Index** |
| 1 | NBA | 0.40 | | 0.85 |
| | NWE | | | 0.82 |
| 2 | NBA | 0.44 | | 0.87 |
| | NWE | | | 0.83 |
| 3 | NBA | 0.38 | 0.23 | 0.73 |
| | NWE | | 0.43 | 0.66 |
| | LWWE | | | 0.81 |
| 4 | NBA | 0.41 | 0.23 | 0.75 |
| | NWE | | 0.38 | 0.66 |
| | LWWE | | | 0.79 |
| 5 | NBA | | 0.20 | 0.73 |
| | LWWE | | | 0.81 |
| 6 | NBA | | 0.24 | 0.76 |
| | LWWE | | | 0.81 |
| 7 | NWE | | 0.41 | 0.78 |
| | LWWE | | | 0.89 |

NBA: number of piglets born alive; NWE: number of weaned piglets; LWWE: litter weight at weaning.

**Table 13.** Genetic correlations between the pure breed and cross breed performances (Pr > |t| < 0.0001).

| Models | Traits | | |
|---|---|---|---|
| | **Traits** | **NWE** | **LWWE** |
| 1 | NBA | 0.50 | |
| 2 | NBA | 0.52 | |
| 3 | NBA | 0.49 | 0.31 |
| | NWE | | 0.59 |
| 4 | NBA | 0.49 | 0.29 |
| | NWE | | 0.56 |
| 5 | NBA | | 0.30 |
| 6 | NBA | | 0.32 |
| 7 | NWE | | 0.58 |

NBA: number of piglets born alive; NWE: number of weaned piglets; LWWE: litter weight at weaning.

**Table 14.** Range of the estimated index scores.

| Models | Group | Scores |
|---|---|---|
| 1 | Large White | −3.97–190.79 |
| | Landrace | 13.37–173.78 |
| | F1 | 27.74–171.60 |
| 2 | Large White | −0.93–188.99 |
| | Landrace | 17.25–175.70 |
| | F1 | 24.88–175.98 |

**Table 14.** *Cont.*

| Models | Group | Scores |
|---|---|---|
| 3 | Large White | −4.15–175.12 |
| | Landrace | 18.79–181.15 |
| | F1 | 14.63–161.33 |
| 4 | Large White | −1.81–178.63 |
| | Landrace | 23.49–178.71 |
| | F1 | 14.66–160.63 |
| 5 | Large White | −2.30–179.89 |
| | Landrace | 18.00–182.61 |
| | F1 | 5.22–163.98 |
| 6 | Large White | −1.03–187.16 |
| | Landrace | 22.74–180.79 |
| | F1 | 6.51–164.31 |
| 7 | Large White | −23.77–172.39 |
| | Landrace | 37.18–168.76 |
| | F1 | 13.14–159.12 |

## 4. Discussion

### 4.1. Heritability, Permanent Environmental Effect

Wolf et al. [14] found in their experiment higher heritabilities for Large White and Landrace breeds for the number of piglets born alive in parity one, which ranged between 0.09 and 0.13 and between 0.09 and 0.12, respectively. For subsequent parities, t higher values (0.10–0.13 and 0.11–0.14) were reported. Kasprzyk [15] found lower values for Landrace breed $h^2 = 0.023$ for NBA, $h^2 = 0.027$ for the number of piglets on the 21st day, and $h^2 = 0.03$ for the litter weight at 21 days. Suárez et al. [16] estimated higher heritability compared to the current experiment for the Large White breed in the frame of 3 parities where estimates were 0.18, 0.17, and 0.19 for NBA, and they obtained the same result for NWE (0.05, 0.07, and 0.05). In the study of Dube et al. [17] for the Large White breed, approximately the same results for heritability traits were found for NBA and NWE with 0.07 and 0.03, respectively. On the other hand, $h^2$ for LWWE was lower compared to that in our research (0.06). Suarez et al. [18] observed two times higher heritability in the Large White breed in the range of 0.15–0.20 for NBA compared to our study, depending on number of parities, but obtained a similar result for NWE (0.03–0.08), depending on parities. Similarly, higher results of heritability were obtained for the Landrace breed for NBA in the range of 0.16–0.27, but NWE had lower heritability, as found in our experiment (0.04–0.09). Nagyné-Kiszlinger et al. [19] got the same result of heritability for NBA as found in our experiment for Large White (0.09), Landrace (0.06), and F1 (0.06–0.07). Krupová et al. [20] found in their study slightly higher results for Large White and Landrace breeds for trait NBA and NWE with values of 0.099, 0.102, and 0.091, 0.076, respectively. Size of heritability is the main factor that affects the future selection process in terms of genetic information as it flows from one generation to the next and directly impacts the diversity in the population of our case pigs. Lower or higher heritability that we observed in different investigations depends on many factors. Firstly, it is nature's limitation for how specific traits contribute to their genetic flow. Secondly, between different herds or the same breeds, genetics can be observed at different stages of the selection process. Thirdly, and maybe the most important factor, it is maintaining proper and precise data collection and future processing of information.

Nagyné-Kiszlinger et al. [19] obtained similar results for permanent environmental effects in NBA with a value of 0.06 for all 3 breeds. Skorupski et al. [21] got the same results for permanent environmental variance ratios of NBA, 0.06 and 0.05 for Large White and Landrace, respectively. Krupa and Wolf [22] found higher results than ours in the case of permanent environmental effects for NWE, with values of 0.05 and 0.06 for Large White and Landrace, respectively. Differences obtained in results for permanent environmental effect can be explained only from one side in that some farms could have been equipped differently in terms of the artificial controlling environment. Therefore, this could be one of the main reasons why the difference is higher or lower. This is also one indicator that should include additional information collected in the classical approach of BLUP methodology. This would allow for a deeper analysis of how different farms are equipped, such as barn temperature fluctuating over the years and how farms were being modernized over a period of time or possibly years

### 4.2. Genetic Trends

The estimated genetic trends for all seven models generally showed the same tendency and varied in a small range. Models 1–2, 3–4, and 5–6 characterized the same traits but partly contained different factors.

Chen et al. [23] estimated different results in contrast to that in the current study for all traits and average values (Large White and Landrace breeds) of genetic trends were 0.018 piglets per year for NBA, 0.114 kg per year for LWWE, and 0.004 piglets per year for NWE. Chansomboon et al. [24] reported negative and significant genetic trends as average for Large White, Landrace, and F1 (cross of Large White and Landrace) for reproduction traits. Genetic trends had the values of −0.017 piglets per year for NBA, −0.019 piglets per year for NWE, and −0.022 kg per year for LWWE. Similarly, in our research, we obtained negative trends but for Landrace breed and F1 cross for NBA and all three traits, respectively. Genetic gain for the Landrace breed in the experiment of Kasprzyk [15] exhibited the same negative tendency as in the current experiment with a value of −0.05 piglets per year for NBA, but in the case of NWE result was different and negative (−0.04 piglets per year) and LWWE had a negative trend of −0.48 kg per year. It should be noted that a breed such as Landrace is selected mainly for growth and carcass traits, which may have contributed to the reported selection inefficiency observed for the analyzed populations. Based on our findings, no substantial genetic improvement can be expected in the near future Hungarian Landrace for our herds.

### 4.3. Genetic Correlation and Index by Breeds

In the work of Suárez et al. [16], negative genetic correlations were found between NBA and NWE in frame of 3 parities respectively (−0.23, −0.96, and 0.01) for the Large White breed in contrast to our findings. However, Dube et al. [17] found positive genetic correlations with values for NBA with NWE and LWWE of 0.88 and 0.32, respectively. NWE and LWWE correlated on level 0.78. Suarez et al. [18] estimated in their study contradictory results for genetic correlation between NWE and NBA depending on the number of parities of the Large White breed (0, −0.66, and −0.67 for parities 1–3, respectively). They obtained similar results for Landrace breed (0.07 and −0.45 for parities 1–2, respectively). On the contrary, Krupová et al. [20] found in their study high correlations between the traits NBA and NWE for Large White and Landrace breeds of 0.954 and 0.979, respectively. The NBA showed a correlation of 0.99 with the NWE and 0.78 with the LWWE in experiment of Kasprzyk [15]. The correlation between NWE and LWWE was 0.82, which was higher compared to that in our study.

### 4.4. Genetic Correlations between the Pure-Bred and Cross-Bred Performances

Based on our results of low to moderate correlation between the pure-bred and cross-bred performances relative to flow inference that increase reproduction performances of cross breed impossible just by using the genetic merit of pure breeds and require just

joined evaluation of genetic merit. Abell et al. [25] concluded, based on their low result of correlation, that selection for longevity or lifetime performance at the nucleus level may not result in improved performance at the cross-bred level. Nakavisut et al. [26] came to the conclusion that the low genetic correlation of traits between pure-bred and cross-bred animals suggests that a joint genetic evaluation may be more appropriate for the genetic improvement of performances for cross-bred pigs.

## 5. Conclusions

The estimated index weighing factors are good tools for construction selection indexes for maternal pig breeds when the desired selection response can be defined. As a result, aggregate genetic merits were constructed; thus, the animals can be selected based on their overall performance.

**Author Contributions:** Conceptualization, O.K., G.K. and I.N.; Data curation, O.K., J.F, G.K. and I.N.; Formal analysis, O.K., H.N.-K., J.F. and I.N.; Funding acquisition, H.N.-K., G.K. and I.N.; Investigation, O.K., H.N.-K., J.F. and I.N.; Methodology, O.K., J.F., G.K. and I.N.; Project administration, G.K. and I.N.; Resources, O.K., G.K. and I.N.; Software, O.K., J.F. and I.N.; Supervision, G.K. and I.N.; Validation, O.K., H.N.-K. and I.N.; Visualization, O.K., H.N.-K. and I.N.; Writing—original draft, O.K. and I.N.; Writing—review and editing, O.K. and I.N. All authors have read and agreed to the published version of the manuscript.

**Funding:** This research is funded by the K128177 (NKFI-6) project.

**Institutional Review Board Statement:** Not applicable.

**Data Availability Statement:** The data presented in this study are contained within this article.

**Acknowledgments:** Tempus public foundation for implementation of Stipendium Hungaricum program.

**Conflicts of Interest:** The authors declare no conflict of interest.

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
