# Peer review of "Genetic Parameters of Reproductive Performances in Hungarian Large White, Landrace, and Their Crossbred F1 Pigs from 2010 to 2018"

_diversity, doi:10.3390/d14121030_

Round 1

Reviewer 1 Report (Previous Reviewer 1)

After the author's revision, the study purpose of this paper has changed and is more consistent with the research content. But there are still some problems that need to be revised.

Title: as the key of animal diversity”, but the inbreeding coefficients, which best reflect genetic diversity, were not analyzed for these pig groups in this study. I suggest that the title should be changed to “Genetic parameters of reproductive performances in Hungarian Large White, Landrace, and their crossbred F1 pigs from 2010 to 2018”

Line 31-36, delete these sentences. Because the global warming and climate change had essentially nothing to do with the genetic diversity of the commercial pig breeds in this study that were raised in artificial environments. The breeding of commercial pig breeds is mainly artificial selection based on market demand, not natural mating and reproduction, and also has nothing to do with the small world inside pig farms.

Line 39-41: “Because we think that the viability of new born generations of pigs or other animals is a guarantee for the future survival of a species or breed.

Line 41-42: “We used in our current research a classical approach for evaluating genetic parameters by applying BLUP methodology”

Line 45, “characterize”

Line 62, delete “are”

Line 71-74, please provide the detail methods for estimation breeding values and variance components using PEST and VCE 6.

Line 99, “Heritability for NBA and NWE was low”

Table 2, How about the “C” represents for? Please add in the Notes.

Table 3, Please add the Number of litters for each pig group in the Table. Change “breed” to “group”. Because the crossbred “F1” generation did not belong to a pig breed.

Table 4, Please add the Method of calculating “Pe”, the Long-term environmental effect, in the Methods. I don't know how you calculated this Pe.

Line 141, “calculated”

Table 9, 10, and 11, What means “Index”? Please provide the p-values for these correlation coefficients. In addition, there was a relationship between NBA, NWE, and LWEE. When calculating the correlation between two variables, it is recommended to use a partial correlation to normalize the third variable for model 3 and 4.

Table 12, Please provide the p-values for these correlation coefficients.

Line 214-217Please rewrite this sentence. Confusion of logical relations

Line 239, “at different stages”

Line 248, “can be explained”

Line 309-312, delete this paragraph. There are no climatic data and conclusions in this study.

Author Response

Many thanks for comments and remarks. Please find attachment.

Reviewer 2 Report (New Reviewer)

The Authors evaluated the genetic parameters, breeding values and aggregate breeding values of number of piglets born alive, number of weaned piglets, and litter weight at weaning in the Hungarian Large White, Hungarian Landrace breeds and in their cross. The genetic merit was calculated using seven repeatability animal models.

Here are some suggestions that the Authors can consider improving the manuscript.

Lines 71-72: please change “BLUP (Best Linear Unbiased Prediction) and REML methodology (Restricted maximum likelihood)” with “Best Linear Unbiased Prediction (BLUP) and Restricted maximum likelihood methodology (REML)”.

Line 144: please change “….were based on so-called…” with “….were based on so called…”.

Lines 214-216: the sentence is confuse and needs to be clarified.

Lines 287-295: the sentences are confuse and need to be clarified. Moreover, I suggest an English revision.

Lines 297-304: In my opinion, this paragraph is part of “Discussion” section.

Author Response

Many thanks for comments and remarks. Please find attachment.

Reviewer 3 Report (New Reviewer)

I would urge the authors to engage the assistance of a colleague with more experience in manuscript preparation and scientific writing - and revise this MS.  It's not clear what the purpose of the study was  or what the results mean. 

Author Response

Many thanks for comments and remarks. Please find attachment.

Reviewer 4 Report (New Reviewer)

Thanks for the efforts from the authors on conducting the analysis. However, there are places overall can be improved in terms of modeling for estimating genetic parameters and the format of presenting. For examples, why not just using a single 3-trait model to estimate the heritability and genetic correlation among traits instead of choosing 7 models (bivariate or 3-traits) but with slightly different options of fixed effects? Also, results from genetic parameters estimations are not well presented herein.

Author Response

Thanks for the efforts from the authors on conducting the analysis. However, there are places overall can be improved in terms of modeling for estimating genetic parameters and the format of presenting. For examples, why not just using a single 3-trait model to estimate the heritability and genetic correlation among traits instead of choosing 7 models (bivariate or 3-traits) but with slightly different options of fixed effects? Also, results from genetic parameters estimations are not well presented herein.

Authors wanted to test different model possibilities. Nevertheless based on the request of the editor using the log likelihood values the best fitting model (model 4) was identified. So if the editor agrees it would also be possible to  present the structure of all the tested models but only provide  the heritability estimates and the genetic correlation coefficients (similarly for the residual and permanent environmental effects) in one table.

Round 2

Reviewer 1 Report (Previous Reviewer 1)

Line 97, please provide the company information of SAS 9.4 software.

References: please check the reference format, for example the 1st, 2nd and 3rd references. 

It is recommended to check the grammar again before publication.

Author Response

Line 97, please provide the company information of SAS 9.4 software.

The requested information was added and SAS is included among the references.

References: please check the reference format, for example the 1st, 2nd and 3rd references. 

References were checked throughout the manuscript. Besides, introduction was re-written as although the temperature is a very important environmental factor, unfortunately within the course of our study we did not have acces to any temperature data.

It is recommended to check the grammar again before publication.

I did my best to read and correct the text also from language aspects.

This manuscript is a resubmission of an earlier submission. The following is a list of the peer review reports and author responses from that submission.

Round 1

Reviewer 1 Report

diversity-1885142-peer-review

Reproductive performance of the Hungarian pig breeds as the key of animal diversity in time of technology and climate changing

The title is very attractive. But when I read it, I found that I was deceived. The content of the article was not closely related to the title. In particular, pigs were raised in an artificial environment in pig barns. The indoor temperature and other environmental conditions were artificially controlled, and the environmental effects of their reproductive traits were only artificial environmental effects, which had nothing to do with the climate changing. The actual content of this manuscript is to evaluate the genetic parameters of the reproductive performance of the Hungary Large White, Landrace and their F1 generation from 2010 to 2018 using 7 different models.

Line 4, delete the period in the title.

Line 62, “reciprocal cross (F1)” including Large White boars mated with Landrace sows and Landrace boars mated with Large White sows?

Line 63, delete “respectively”

Line 71-73, delete the repeated “using the PEST…REML method.”

Line 87, “various software”, please specify the name and edition of the software.

Line 152-154“1850 to 1900 yBecause”, years? Please delete the last repeated sentence.

Line 278-286, Discussion 4.5 “BLUP methodology relation to technological and climate chenges in years”. “chenges” should be “changes”. According to the title, this part should be the most important in this study. But the authors just raise the question. Permanent environmental effect derived from the experimental information “just for environment inside farms”. Therefore, this study is not at all relevant to the title and the topic of Diversity journal.